# Energy Effectiveness of Jet Fuel Utilization in Brazilian Air Transport

**Manoela Cabo** [1,2,*]👤, **Elton Fernandes** [2]👤, **Paulo Alonso** [3]👤, **Ricardo Rodrigues Pacheco** [2]👤
**and Felipe Fagundes** [2]

[1] IBGE Brazilian Institute of Geography and Statistics, Av. República do Chile, 500-Centro,
Rio de Janeiro-RJ 20031-170, Brazil

[2] COPPE Production and Transport Engineering Program, Federal University of Rio de Janeiro, Av. Pedro
Calmon, 550-Cidade Universitária da Universidade Federal do Rio de Janeiro,
Rio de Janeiro-RJ 21941-901, Brazil; tglcoppe@gmail.com (E.F.); rpacheco48@gmail.com (R.R.P.);
felipefagundes.jf@gmail.com (F.F.)

[3] Electronics and Communication Engineering Department, UERJ Rio de Janeiro State University, R. São
Francisco Xavier, 524-Maracanã, Rio de Janeiro-RJ 20550-000, Brazil; alonso.psr@gmail.com

[*] Correspondence: mcabo@impa.br; Tel.: +55-21-98206-3000

**Abstract:** Since World War I, the commercial aviation industry has seen many improvements that now allow people and goods to reach the other side of the world in a few hours, consuming much less fuel than in recent decades. Improvements in cargo capacity and energy efficiency were significant, and in this scenario, commercial airlines were able to thrive and bring great benefits to the world economy. However, this sector is facing environmental challenges due to the intensive use of jet fuel. Brazil is one of the largest domestic air passenger markets in the world and still has great growth potential, considering its economic potential and territorial dimensions, which are roughly the same size as the US and twice the size of the European Union. This paper discusses the partial productivity of jet fuel in Brazilian domestic aviation and proposes an econometric method to support public regulators and airline decisions. The proposed model uses variables, such as aircraft size, route characteristics, and idle flight capacity, in a panel data analysis. The results show that reducing idle capacity is one of the best ways to achieve better short-term fuel efficiency, and therefore will reduce the environmental impacts and have positive economic effects on commercial air transport activities. This paper brings a new approach to the discussion of airline performance, focusing on the use of jet fuel, with economic and environmental consequences.

**Keywords:** jet fuel productivity; idle capacity; air transport; Brazil

## 1. Introduction

Research to improve jet fuel usage has been an ongoing effort for aviation companies and mainly airlines and turbines manufactures. Discussions on fuel consumption efficiency can help sustainable air transport activities around the world, and particularly in emerging countries, such as Brazil.

The aviation industry is at the forefront of industry when it comes to reducing the adverse impacts of its activities, which to some extent contribute to the economic efficiency and sustainability of air transport services. Market pressures, including regulatory measures and competition, require business productivity improvements, whether they are material suppliers (such as turbine and aircraft manufacturers), service providers (such as air navigation service providers), ground operators (such as airports), or the major entities in the production chain, the airlines themselves [1]. To some extent, the environmental and financial objectives of airlines are convergent with the airline's main

operating cost [2] and the idle capacity is a concern in any industry because it could result in increased capital needs and operating costs.

However, Schnell [3] suggests that airlines intentionally maintain idle resources to respond to uncertain demand, anticipated growth opportunities, and competition. This happens despite affecting the cost position of airlines by increasing the main operating cost component, which is fuel. In addition, environmental issues are on the agenda of all sectors. An important measure of efficiency in air transport is the weight carried by burnt fuel. This article discusses how to manage factors that affect partial jet fuel productivity and how to achieve a situation that will allow the air transport industry to sustainably improve. Fuel burn productivity can be influenced by the airline's idle capacity as expressed by the unused portion of the aircraft capacity (load factor), aircraft size, and weekly frequency, among other factors.

When an airline decides to take a specific route, it must make decisions about which aircraft to use and how often to fly a specific route. It also makes decisions about which aircraft model to use and its flight frequency. In this study, all variables were indicated by their annual averages on each route. Givoni and Rietveld [4] found that airlines' choice of aircraft size depends on the route characteristics rather than airport characteristics. Accordingly, aircraft size and flight frequency can be seen as representing the supply side of the route characteristics.

Although air transport is not on the list of the activities that most pollute the planet, its main environmental effect occurs at high altitudes, a very specific location and one that is of major concern to environmentalists [5]. As a result, because of their possible adverse impacts, air transport activities have been monitored with considerable attention.

In Brazil, fuel burn in domestic air transport constitutes the leading source of pollution by this sector. In that light, this paper examines how air transport performance has progressed in terms of its consumption of jet fuel and discusses the prospects for the future. An unsolved problem is to determine whether there is room to improve performance with existing technologies. Observation of real data in a market like Brazil's, estimated to become the fourth largest national market by the late 2020s [6], can yield important information regarding the world scenario. The paper's scientific contribution is concentrated on using an objective method to ascertain performance trend patterns in an industry that is important to the issue of sustainability and thus opening up a discussion on possible sector measures that can lead to new productivity gains, which are an essential factor for an industry with prospects of high growth rates in coming years.

*Literature Review*

Fuel burn correlates strongly with emissions, thus contributing directly to undesirable externalities in air transport [7] and it is the airlines' main cost item [2]. Papers examining issues relating to jet fuel burn generally address airlines' efficiency and their pollution potential, namely the depletion of oil and the possibility of replacing the fuel currently used with another that is less polluting, pollution-free, or renewable, the option of other means of transport, or even the introduction of an additional tax on jet fuel burn. Although no extensive literature review has been presented, studies considered relevant to defining the approach of this paper were selected.

Simões and Schaeffer [8] examined Brazilian air transport's contribution to greenhouse gas emissions. They offer a series of mitigation options, which include improving air traffic flow management, introducing a tax on each flight based on the jet fuel burn for the route, introducing intermodal options for the dense air connections between Rio de Janeiro and São Paulo by implementing a high-speed train, and so on. They also estimate that, if the recommendations they propose were applied, they would reduce long-term $CO_2$ emissions from air transport by 28.5%.

Chèze et al. [9] made projections for jet fuel demand in eight regions of the world for the period 2008–2025. Their forecasts were based on an econometric model using dynamic panel data. The main scenario they explored involved a 100% increase in world air traffic in the period, at an annual growth rate of 4.7%. In that same period, fuel burn would grow by 38%, at an annual 1.9%. They claim that this

growth in consumption already takes account of all efficiency improvements to turbines and aircraft aerodynamics. Accordingly, their opinion is that technological progress will continue to be essential to mitigating the impacts of increasing air traffic on fuel burn. Given that the improvements introduced have not been reflected in lower fuel burn rates, they alert that it will take industry-wide disruptive innovation for that scenario to change.

O'Kelly [10] examined the efficiency of hubs from the environmental standpoint using fuel burn as an indicator of environmental cost. He considered the fuel cost associated with larger aircrafts in order to determine the implications of high load factors on dense routes and thus to specify the implications for hub and gateway location. He showed that by adding a fixed charge when modelling fuel burn, a multiple-allocation hub-and-spoke model can be adjusted to direct flow to the inter-facility connector. Chang et al. [11] studied the economic and environmental efficiency of 27 global airlines in 2010 using a data envelopment analysis model with the weak disposability assumption. They concluded that Asian airlines are generally more efficient, while the operational and environmental performance of European and North American airlines are inefficient. Airlines' inefficiency can be attributed to two main factors: inefficient fuel burn and a less diversified revenue structure.

Park and O'Kelly [12] estimated fuel burn by considering the distance between markets, with a given aircraft fleet composition and seat configuration. They concluded that distance is a crucial factor in estimating fuel burn: on long-haul routes, the lowest fuel burn rates are found in operations from 1000 to 2500 nautical miles. They speculated that fuel burn per seat-distance can be considered a criterion for levying environmental taxes on airlines. Zou et al. [13] investigated fuel efficiency among 15 main airlines and their subsidiaries in the United States. They found that fuel burn is amply explained by, and strongly correlated with, revenue passenger miles (RPMs) and the number of take-offs, and that although regional airlines have improved their accessibility provision, they display a higher fuel burn per RPM.

González and Hosoda [14] examined the growing impact of commercial aviation on $CO_2$ emissions in Japan, as well as the potential impact on climate change. The investigation comprised the effects of the jet fuel tax introduced by the Japanese government on all domestic flights. They used a time series model with monthly observations of fuel burn from 2004 to 2013. They estimated the amount of $CO_2$ emissions that would be produced in the event the fuel tax was not applied. Cui et al. [15], using data envelopment analysis, studied the impacts on airline performance resulting from emissions limits established by the European Union. They used total revenue as a desirable output and greenhouse gas emissions as an undesirable output. As input variables, they used the number of employees and jet fuel burn. The sample comprised 18 large global airlines, from 2008 to 2014, and the findings where in terms of which airlines showed the most potential for increasing their outputs. Cui and Li [16] used data envelopment analysis to measure the dynamic efficiency of 19 airlines from 2009 to 2014. The input variables selected were the number of employees and jet fuel burn, while the outputs were revenue tonne-kilometers, revenue passenger kilometers, and total revenue. The dynamic factor selected was capital stock. They concluded that Scandinavian, Emirates, and Cathay Pacific were the sample's benchmark airlines.

Zou and Chau [17] estimated fuel price effects on freight volumes for various modes of transport in Shanghai. They found a causality running from rail to road transport where allocating more time and routes for rail freight traffic and reducing rail freight taxes can increase the volume of rail freight and thus decrease overall energy use. Their findings contribute to the economics of freight transport and correlate with the intent of our paper, which aimed to show that idle capacity utilization can bring fuel efficiency without increasing gas emissions.

It can be seen from the literature review that research into jet fuel burn is directed toward the environmental issue and leaves aside the discussion of financial benefits to airlines. There is a trend for such studies to address the issue of fuel burn in association with the definition of a benchmark company and its relationship with the environment. Mostly, there is a certain pessimism as to the possibility of significantly reducing fuel burn through short- to medium-term technological advances. However, there is the argument of the need for disruptive innovation in order to attain new levels of

efficiency. The literature reviewed offered no discussion of operational proceedings that airlines in general can introduce on routes to reduce jet fuel burn. This study also examined the mean partial productivity of fuel burn via air routes, regardless of the airline, in terms of idle capacity and other market characteristics in Brazilian regular domestic air transport. This knowledge will both assist airlines in their planning and operating procedures, and inform the government formulation of regulations and policies to stimulate air transport, all with due regard for environmental concerns.

Therefore, the main goal of this study was to examine jet fuel productivity as an important element in business costing and emissions. In developing countries, such as Brazil, where air transport is a small part of the transport matrix, the focus is not only on reducing emissions but also to address the responsible use of jet fuel, with idle capacity being a fundamental element for its productivity. Therefore, this work will be limited to the analysis of operational alternatives that could contribute to fuel productivity, as well as proposing short-term solutions to reduce the environmental impacts caused by the burning of jet fuel.

## 2. Materials and Methods

The choice of methods was based on the purpose of the research and the availability of information. In Brazil, there is no long-term time series information about air links. Another element is that jet fuel data is available only for domestic air companies. The panel data approach proved to be adequate because it has cross-section and periodic dependences that have specific characteristics. Initially, tests were performed by considering only one stage; however, the possibility of endogeneity in the variables was observed, and thus, the two-stage method with an instrumental variable was adequate. In addition, the statistical tests recommended the adoption of fixed effects, both in the cross-section and in the period. The choice of variables was the result of a search, even in other sectors of activity [3], which provided information related to the company's operating activities.

This research was developed following the steps shown in Figure 1. The first step was the motivation for this research and the second was to find a key performance indicator to measure jet fuel efficiency. In the third step, data that could be linked to jet fuel efficiency was collected and processed, available data were searched for, and the necessity of their preparation was verified. The fourth step was the effective preparation of the data. In the fifth step, the unbalanced panel data model was analyzed to understand which factors within the collected data could influence jet fuel efficiency. The sixth, seventh, and eighth steps involved the choice of the analytical methodology to be adopted and its results. For the final target, this research demonstrated that a reduction in the idle capacity for air transport can significantly improve environmental results.

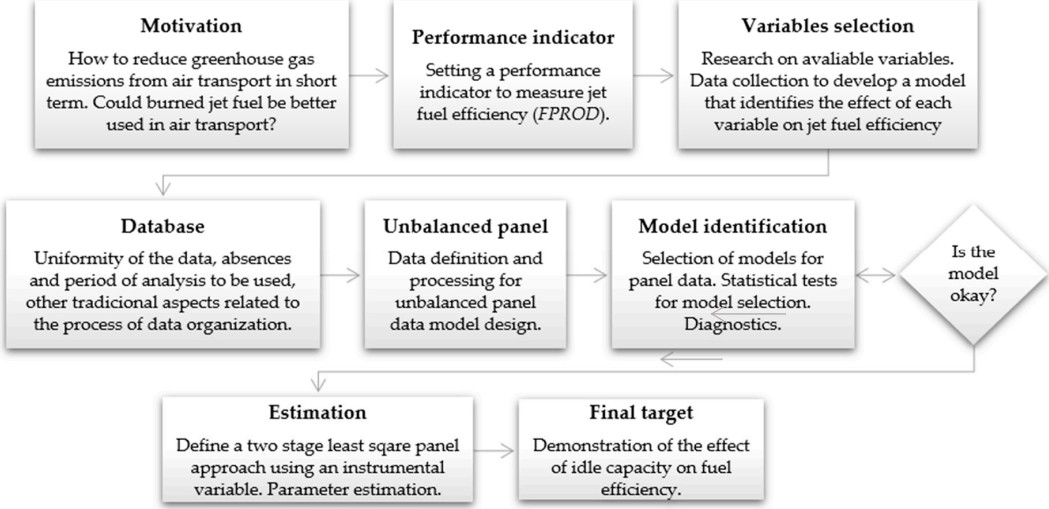

**Figure 1.** Flowchart that describes the steps developed in the research.

*2.1. Analytical Methodology*

　　Panel data is a structure that is recommended when the explanatory variables are time-dependent, also known as longitudinal data, representing repeated observations of a set of units in a cross-section. That is, the predictive and explanatory variables of interest are measured on different occasions, generally over time, for each single individual or element (in the case of this study, air routes). In longitudinal studies, the observations of an individual over time are correlated and thus demand statistical techniques that take account of that dependence [18]. Longitudinal data offer several advantages over data distributed over a cross-section or time series only. The benefits include being able to study dynamic relations over time and model differences among individuals [19]. Approaches using econometric analysis of panel data have evolved over the years and experts have developed several methodologies to contemplate specific characteristics of the observed data [20]. The literature recommends experimenting with various approaches in order to select the most appropriate modelling. Statistical tests have been developed to assist the process of selecting among approaches.

　　The analytical model proposed in this paper endeavored to explain the relationship between partial productivity of fuel and idle capacity. As the study focused on average air transport efficiency per route per year, no distinction was made between airlines and the aspects of competition among airlines on the routes were not addressed. It is important to point out that in the model addressed in this study, we aimed to explain a variable related to the operational cost of the route that is strongly linked to the fuel consumption. Thus, the productivity of the fuel in a specific route in a given year can be determined by operational characteristics of the airline company operating that route in that specific year. There is an expectation that with the passing of years, we can observe an improvement in the performance of airline companies related to fuel usage, either by the refinement of operational procedures or by continuous technological improvement in the industrial sector. In this sense, the hypothesis of a fixed or random annual effect was tested in the choice of the model. Another important aspect is the characteristics of each route that cannot be explained solely by the distance between two airports. It is necessary to consider an effect for each route that must be fixed or random, with the appropriate statistical tests carried out. Therefore, in order to confirm the suitable type of model, we will performed the following tests: redundant fixed effects or Chow test (likelihood ratio), omitted random effects (Lagrange multiplier), and correlated random effects (Hausman test) [19].

　　Once the panel data approach is defined, it is necessary to consider the possibility of endogeneity in the model, which can define a two-stage least squares panel approach using an instrumental variable. Therefore, the model included: a variable that represents the usage level of the airplanes (capacity), an operational variable defining the airline companies in the route, and an instrumental variable that could mitigate endogenous problems in the model. Other variables, such as waiting time and aircraft taxi data, also affect the partial fuel productivity, but no data were available to estimate their impact. For a thorough discussion of the two-stage least squares panel approach, see Hsiao [20], and for software implementation, see EViews 11 [21]. The dependent variable was the tonne·km transported per liter of fuel. The variable linked to the level of usage was the idle capacity, which was determined using the airplane's total capacity minus the annual average load factor. An airline company's operational decision regarding the operation of the route will be represented by the annual average payload offered on that particular route. The model's instrumental variable was the average weekly frequency of flights in a specific route per year. The proper effects of each route and of each period were defined in accordance with the results suggested by the statistical tests carried out (fixed or random effects). In the period considered, it a uniform pattern of the fleet usage was observed for the routes considered. Once the analysis was performed for each route per year, the specification and technical characteristics of the airplanes were not included in the formulation of the models. The software used to perform the regressions was the econometric software EViews 11 [21], which is a statistical modeling software. In general, for processing, there is no strict specification about the hardware as the model does not require much computational effort and can be processed on a personal computer. In this research, an Intel Core i7 computer with 16 GB RAM memory was used.

The equations' notation was in accordance with EViews 11 [21]; however, they were adapted for the study variables. All variables reflect the annual mean on each route (from city $i$ to city $j$) for each year $t$. The corresponding model used to estimate the regression parameters is shown in Equation (1).

$$\ln FPROD_{i,j,t} = c + \omega_{i,j} + \eta_t + \alpha \overline{lnIC}_{i,j,t} + \beta \ln ASIZE_{i,j,t} + \varepsilon_{i,j,t} \tag{1}$$

where

$c$: constant
$ln$: the natural logarithm of the variables;
$\omega_{i,j}$: estimated cross-section fixed effect coefficients;
$\eta_{i,j}$: estimated period fixed effect;
$\alpha$ and $\beta$: estimated regression model coefficients;
$FPROD_{i,j,t}$: mean fuel productivity on route $i$ to $j$ in year $t$ (tonne·km/L);
$ASIZE_{i,j,t}$: mean aircraft size on route $i$ to $j$ in year $t$ (kg);
$\overline{IC}_{i,j,t}$: mean idle capacity on route $i$ to $j$ in year $t$ (ratio);
$\varepsilon_{i,j,t}$: regression error.
$\overline{IC}_{i,j,t}$ is the instrumental variable, as given in Equation (2):

$$\overline{lnIC}_{i,j,t} = c + a \ln ASIZE_{i,j,t} + b \ln WF_{i,j,t} + u_{i,j,t}, \tag{2}$$

where

$WF_{i,j,t}$: mean weekly frequency on route $i$ to $j$ in year $t$;
$u_{i,j,t}$: regression error;
$c$, $a$, and $b$: estimated regression model coefficients.

## 2.2. Data

The data set was formatted as unbalanced panel data for domestic air routes in Brazil from 2007 to 2016. Although the database comprised information since 2000, the year 2007 was chosen for the analysis by considering the years when the four airline companies operating the domestic Brazilian air routes reached 90% of market share (see Table 1). From 2000 up until 2006, Brazil experienced a process of consolidation and bankruptcy of Brazilian airline companies, which caused the market to be unstable in terms of its operational conditions. From 2007 on, two airline companies, TAM and GOL Airlines, dominated the Brazilian domestic air market. After this year, AVIANCA Airlines, which entered the market in 2003, began to have a significant share; in 2008, AZUL Airlines entered the market, very quickly increasing its share. In 2016, these four companies represented 99% of the market share for commercial domestic routes. The evolution of participation in revenue passenger-kilometers (RPKs) in the total of Brazil is shown in Table 1. As the structure of the Brazilian domestic air transport is almost the same in 2019, and there were no substantial changes in the fleet of aircrafts, we believe that the research results are still valid.

The annual information was organized using the available Brazilian National Civil Aviation Agency (ANAC) database. The main reason the data set was unbalanced is the variation in Brazil's air transport network during the study period, especially the regional routes. In the case of missing data, the unbalanced panel considered all underlying information, such as years, not excluding the cross-section by total. Additionally, to avoid the presence of outlier data, two conditions were stablished for the sample. The first considered that the route should have at least an average of one round trip per week, this filter selected the regular routes, removing any outlier from the base. The second established that the load factor should be higher than 10%. These two conditions limited the sample to regular operations throughout the year, avoiding seasonal or sporadic operations.

**Table 1.** The four main airlines' percentage of the Brazil's total domestic RPK.

| ANO | TAM | GOL | AZUL | AVIANCA | TOTAL |
|-----|-----|-----|------|---------|-------|
| 2000 | 14% | 0% | | | 14% |
| 2001 | 30% | 5% | | | 35% |
| 2002 | 34% | 11% | | | 45% |
| 2003 | 32% | 19% | | 0% | 51% |
| 2004 | 35% | 21% | | 0% | 56% |
| 2005 | 42% | 26% | | 0% | 68% |
| 2006 | 48% | 34% | | 1% | 84% |
| 2007 | 48% | 40% | | 2% | 90% |
| 2008 | 50% | 37% | 0% | 3% | 90% |
| 2009 | 45% | 41% | 4% | 3% | 92% |
| 2010 | 43% | 40% | 6% | 3% | 91% |
| 2011 | 40% | 37% | 9% | 3% | 89% |
| 2012 | 40% | 34% | 10% | 5% | 90% |
| 2013 | 40% | 35% | 13% | 7% | 95% |
| 2014 | 38% | 36% | 17% | 8% | 99% |
| 2015 | 37% | 36% | 17% | 9% | 99% |
| 2016 | 35% | 36% | 17% | 11% | 99% |

Once there was a relation linking the amount of fuel an airplane needed at the moment of take-off, the total weight of the airplane, the embarked weight, and the destination airport, an approach to measure the partial productivity of fuels was the transported weight per fuel unity because this ratio reveals the specific average performance of fuel usage, which is an important input for airline companies. Thus, in this study, we adopted the work load unit (WLU) as an indicator of the weight being transported. Fuel is the most relevant item in the increasing of airline companies' operational costs in Brazil. Estimates indicate that 40% of those total operational costs are due to the fuel consumption.

Fuel efficiency was chosen as the dependent variable because it displays characteristics that are important both to airline performance and to monitoring the use of this resource and related environmental impacts. Airlines reduce their operating costs by increasing the productivity of this important item on their cost spreadsheets. Meanwhile, society benefits from an activity that is essential to economic and social development through its efficient use for which there is, as yet, no alternative, but which results in adverse environmental impacts. Another no less important aspect is that productivity is fundamental to economic and social development, and no measure should neglect this variable as it is an important item in the decision-making process in terms of both air transport policy and airline operations planning. At present, one prominent policy aspect is the restriction on pollutant gas emission levels. The fuel productivity variable is expressed by Equation (3):

$$FPROD_{i,j,t} = \frac{RTK_{i,j,t}}{FUEL_{i,j,t}}, \tag{3}$$

where

$RTK_{i,j,t}$: total revenue tonne-kilometers on route $i$ to $j$ in year $t$;
$FUEL_{i,j,t}$: total fuel burn on route $i$ to $j$ in year $t$ (liter).

A revenue tonne-kilometre ($RTK$) is generated when a metric tonne of revenue load is carried one kilometer. Where that load includes a passenger load, the number of passengers is converted into a weight load, usually by multiplying this number by 90 kg (to include baggage). This figure is based on Doganis [22], which reports that most airlines use 90 kg to express a passenger and baggage, so 11,111 passengers are equivalent to 1 tonne.

The independent variable of the model, the idle capacity $(1 - LF_{i,j,t})$, represents what percentage of capacity offered is not used by the market. The load factor variable ($LF_{i,j,t}$) is estimated as the ratio of

*RTK* to *ATK*. The available tonne-kilometers (*ATK*) is the volume of tonne-kilometers offered, that is, the sum of the product of payload (the total available load weight per aircraft available for transporting passengers, freight, and post) and the route distance. A high load factor means that the flight is being more fully utilized, and accordingly, it is expected that fuel burn productivity will be higher. This is one of the main indicators of air transport performance and will be significantly related to the partial productivity of fuel. The variable $LF_{i,j,t}$ is expressed using Equation (4):

$$LF_{i,j,t} = \frac{RTK_{i,j,t}}{ATK_{i,j,t}}, \tag{4}$$

where

$RTK_{i,j,t}$: total revenue tonne-kilometers on route *i* to *j* in year *t*;
$ATK_{i,j,t}$: total available tonne-kilometers (supplied) on route *i* to *j* in year *t*.

The aircraft size variable $\left(ASIZE_{i,j,t}\right)$ is represented using the mean payload supplied on the route in a certain year. The variable $ASIZE_{i,j,t}$ (in kg) it is expressed using Equation (5):

$$ASIZE_{i,j,t} = \frac{PAYLOAD_{i,j,t}}{TAKEOFFS_{i,j,t}}, \tag{5}$$

where

$PAYLOAD_{i,j,t}$: total payload supplied on route *i* to *j* in year *t* (kg);
$TAKEOFFS_{i,j,t}$: total take-offs on route *i* to *j* in year *t*.

This is a decisive variable through which the airline determines how much transport capacity to offer on the market. Although cases of over-supply may exist for reasons of competition, such cases are distributed across all operations on the route in the year, thus reducing the bias that such cases can cause in assessing the variable. As the study worked with a very large data set, it is to be expected that such distortions will be minimized.

As the presented model suggests the possibility of endogeneity, it was necessary to include a decision variable related to the airline companies in order to mitigate this problem. This was done using a two-stage least squares panel model estimation. The chosen variable was the average weekly frequency of take-offs observed in a specific year for that route. This variable is defined according to Equation (6):

$$WF_{i,j,t} = \frac{TAKEOFFS_{i,j,t}}{52}, \tag{6}$$

where

$WF_{i,j,t}$: average weekly frequency on route *i* to *j* in year *t*.

## 3. Results

### 3.1. Case Study

Brazil covers a geographical area of 8.5 million km², making it the world's fifth largest country, after Russia, Canada, China, and the United States. After Brazil, the countries with the largest territories are Australia and India. It also has the fifth-largest population: 205 million in 2015. Its economy ranks seventh, with a gross domestic product (GDP) of about 3.192 trillion Intl$ (PPP $—International dollar) in 2015. However, also in 2015, with a per capita GDP of about 15,600 Intl$, Brazil ranked only 76th, far from the group classified as developed countries, as is also the case with other countries with large territories and significant GDPs, such as Russia, China, and India. Brazil is a federal republic of 26 states and a federal district (the capital city). Each Brazilian state has a municipality where the state capital city is located. Brazil has 5570 municipalities, of which, 309 had populations of more than 100,000

residents in 2016, a condition that may be considered attractive for air transport operations when the town is at a certain distance from any airport with regular air transport. Rail passenger transport is practically non-existent, and the road transport network is weak in much of Brazil, particularly in the mid-western and northern regions, and in a large part of the northeast. Figure 2a–c shows the number of domestic embarked passengers, amount of cargo embarked at Brazilian airports, and the number of towns served by regular domestic air transport, respectively. Domestic air transport accounts for 90% of the passengers embarked at Brazil's airports, which was just over 85 million passengers in 2016. Meanwhile, it is important to note that, from 2015 to 2016, the number declined as a result of the severe political and financial crisis affecting Brazil. Domestic cargo grew between 2007 and 2011 but has subsequently steadily declined. The number of airports offering regular domestic air transport services has decreased over the years, indicating that traffic is being directed toward higher-density routes where it is easier to optimize operations.

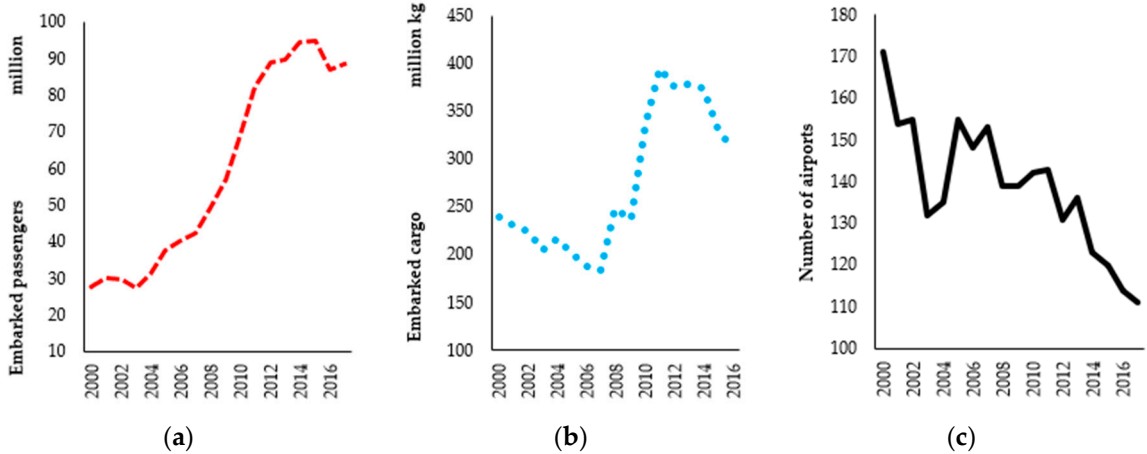

**Figure 2.** (**a**) Domestic embarked passengers in million, (**b**) domestic embarked cargo movement in million (kg), and (**c**) number of airports. Source: Brazilian National Civil Aviation Agency (ANAC).

Table 2 shows the passenger aircraft fleets of the Brazilian air transport airlines for a set of years. The figures in the table show a decreasing number of smaller aircraft, while the number of those with more seats are growing, a trend that is significant when thinking about Brazilian regular air transport. The two first levels of Table 2 show significant changes throughout the years and are linked to an important reduction of the regional routes in the period. We can see that in the case of regional routes, there is a trend toward standardization of airplanes in the range of 51–150 seats. In the three intermediary levels, we see a diminishing of the number of airplanes due to a search for optimization in domestic traffic related to regular and non-regular main routes. The two lowest levels of airplanes are related to long-haul international routes, which are not addressed in this paper.

**Table 2.** Aircraft fleet of Brazilian airlines in 2000, 2007, and 2016.

| Seats | 2000 | 2007 | 2016 |
| --- | --- | --- | --- |
| Up to 50 | 141 | 122 | 6 |
| 51–100 | - | 36 | 52 |
| 101–150 | 212 | 182 | 139 |
| 151–200 | 13 | 226 | 211 |
| 201–250 | 13 | 50 | 48 |
| 251–300 | 13 | 14 | 5 |
| Over 300 | - | - | 16 |
| Total | 405 | 630 | 498 |

Source: ANAC.

The movement is consistent with efforts by airlines, which are undergoing financial difficulties and have not returned positive financial statement balances in recent years, to optimize their operations. This trend reduces the flexibility available to airlines for matching supply and demand, particularly for regional aviation routes where demand is fluctuating.

Tables 3 and 4 show, respectively, the descriptive statistics and correlation matrix of the variables used in the analysis in terms of the natural logarithm and their specific units. By and large, all the variables can be seen to vary substantially about the mean, where the upper and lower values are quite distant. de Neufville [23] argues that short distances in air transport increases the effective fares. In the case of Brazil, these increases make the connection very irregular and not sustainable. Although arbitrary, the minimum flight distance considered for analysis was 200 km, which is regarded as reasonable by authors for establishing regular flights between two localities in this country.

**Table 3.** Descriptive statistics for variables of the model.

| Variables | $ln\,(FPROD_{i,j,t})$ | $ln\,(IC_{i,j,t})$ | $ln\,(ASIZE_{i,j,t})$ | $ln\,(WF_{i,j,t})$ |
|---|---|---|---|---|
| Upper | 2.82 | −0.22 | 10.47 | 6.57 |
| Lower | −4.13 | −4.30 | 7.59 | 0.00 |
| Mean | 0.50 | −0.90 | 9.30 | 2.47 |
| Standard deviation | 0.43 | 0.31 | 0.51 | 1.22 |
| Observations | | 5839 | | |
| **Correlation Matrix** | | | | |
| $ln\,(FPROD_{i,j,t})$ | 1 | | | |
| $ln\,(IC_{i,j,t})$ | −0.70 | 1 | | |
| $ln\,(ASIZE_{i,j,t})$ | 0.52 | −0.31 | 1 | |
| $ln\,(WF_{i,j,t})$ | 0.22 | −0.23 | 0.53 | 1 |

**Table 4.** Descriptive statistics for variables of the model without the logarithm.

| Variables | $FPROD_{i,j,t}$ | $IC_{i,j,t}$ | $ASIZE_{i,j,t}$ | $WF_{i,j,t}$ |
|---|---|---|---|---|
| Upper | 16.80 | 0.80 | 35,100 | 714 |
| Lower | 0.02 | 0.01 | 1980 | 1 |
| Mean | 1.79 | 0.43 | 12,246 | 26 |
| Standard deviation | 0.73 | 0.12 | 5165 | 50 |
| Observations | | 5839 | | |
| **Correlation Matrix** | | | | |
| $FPROD_{i,j,t}$. | 1 | | | |
| $IC_{i,j,t}$ | −0.69 | 1 | | |
| $ASIZE_{i,j,t}$ | 0.49 | −0.31 | 1 | |
| $WF_{i,j,t}$ | 0.027 | −0.08 | 0.31 | 1 |

In the period 2007–2016, the means of all variables had a favorable variation for the improvement of performance (Table 5). Fuel productivity improved in regular domestic aviation in Brazil. From 2007 to 2016, the annual mean *RTK* per liter of fuel observed in the sample increased from 1.95 to 2.40. The annual mean of *IC* reduced from 0.42% to 0.34%, the annual mean of *ASIZE* reduced from 17,310 kg to 16,483 kg, and the annual mean of *WF* increased from 8841 to 15,087 per week.

**Table 5.** Annual evolution of the annual mean of *FPROD, IC, ASIZE,* and *WF* from 2007 to 2016. *RTK*: revenue tonne-kilometers.

| Year | *FPROD* (*RTK/L*) | *IC* (%) | *ASIZE* (kg) | *WF* (Per Week) |
|------|------|------|------|------|
| 2007 | 1.95 | 0.42 | 17,310 | 8841 |
| 2008 | 1.95 | 0.41 | 17,215 | 9231 |
| 2009 | 1.96 | 0.34 | 14,892 | 10,697 |
| 2010 | 2.11 | 0.36 | 16,458 | 12,208 |
| 2011 | 2.19 | 0.35 | 16,484 | 13,676 |
| 2012 | 2.19 | 0.34 | 16,359 | 14,352 |
| 2013 | 2.19 | 0.33 | 16,344 | 15,169 |
| 2014 | 2.30 | 0.31 | 15,491 | 17,172 |
| 2015 | 2.29 | 0.34 | 16,080 | 16,942 |
| 2016 | 2.40 | 0.34 | 16,483 | 15,087 |

*3.2. Panel Analysis*

As described in the Material and Methods section, the Chow, Breusch-Pagan, and Hausman tests were applied for the definition of the estimation pool, with fixed or random effects for the cross-sections and the periods. The Chow test (Table 6) rejected (*p*-value < 0.05) the null hypothesis of the pool estimation for the cross-sections and the periods, suggesting that the fixed effects were more suitable.

**Table 6.** Redundant fixed-effects tests (Chow test).

| Test Cross-Section and Period Fixed Effects | | | |
|------|------|------|------|
| **Effects Test** | **Statistic** | **d.f.** | ***p*-Value** |
| Cross-section F | 9.91 | (987) | 0.00 |
| Period F | 21.18 | (9) | 0.00 |
| Cross-Section/Period F | 10.22 | (996) | 0.00 |

The tests to verify the random effects versus no effect (pool) also rejected the null hypothesis of the pool modelling (*p*-value < 0.05), suggesting in this case that the random effect was the most suitable. Table 7 shows the results of the tests performed.

**Table 7.** Lagrange multiplier tests for random effects (Breusch-Pagan Lagrange multiplier).

| | Test Hypothesis | | |
|------|------|------|------|
| | **Cross-Section** | **Time** | **Both** |
| Breusch-Pagan | 4624.83 | 811.51 | 5436.34 |
| *p*-value | (0.00) | (0.00) | (0.00) |

For the definition between the fixed and random effects for the cross-sections and the periods, the Hausman Test was applied (Table 8). For both the cross-sections and the periods, the Hausman test rejected the null hypothesis for random effects (*p*-value < 0.05). In this case, the modeling developed considered the fixed effects for cross-sections and for periods.

**Table 8.** Correlated random effects using the Hausman test.

| Test Summary | $\chi^2$ **Statistic** | $\chi^2$ **d.f.** | ***p*-Value** |
|------|------|------|------|
| Cross-section random | 304.02 | 2 | 0.00 |
| Period random | 8.45 | 2 | 0.01 |

After examining the various possible options for evaluating panel data modeling, a two-stage least squares panel regression model was selected as the most suitable estimator for this analysis. The Chow test (for cross-section fixed effects) indicated that the fixed-effect model fitted better than the pool approach. On the same way, Breusch-Pagan Lagrange multiplier test indicated that random fixed effects fitted better than the pool approach. Finally, the Hausman test (for cross-section random effects) showed that the fixed effect was also better suited than the random-effect for both the cross-section and period. Table 9 shows the result of the regression model applied in the study.

**Table 9.** Two-stage least squares panel regression model.

| Dependent Variable: $ln\ (FPROD_{i,j,t})$ | | | | |
|---|---|---|---|---|
| Periods Included: 10; Cross-Sections Included: 988 | | | | |
| Total Panel (Unbalanced) Observations: 5839 | | | | |
| Instrument Specification: $ln\ (ASIZE_{i,j,t})$, $ln\ (WF_{i,j,t})$, and C | | | | |
| **Variable** | **Coefficient** | **Std. Error** | **t-Statistic** | **Prob.** |
| $ln\ (\overline{IC}_{i,j,t})$ | −0.94 | 0.47 | −2.01 | 0.04 |
| $ln\ (ASIZE_{i,j,t})$ | 0.48 | 0.15 | 3.24 | 0.00 |
| C | −4.81 | 1.80 | −2.67 | 0.01 |
| **Effects Specification** | | | | |
| | Cross-section fixed (dummy variables) | | | |
| | Period fixed (dummy variables) | | | |
| $R^2$ | 0.84 | | | |
| Adjusted $R^2$ | 0.81 | | | |

## 4. Discussion

The coefficients of the explanatory variables (Table 6) represent the constant elasticities of each of them in relation to a dependent variable. The constant elasticity hypothesis can be considered a limitation of the model. However, it is reasonable in that the operating technology in question is similar in all the airlines and no technology breakthrough was observed in the study period. This was confirmed to some extent by the high level of significance of the coefficients of the independent variables. The elasticity of $IC_{i,j,t}$ (−0.94) indicated the negative impact that idle capacity had over fuel productivity; furthermore, the elasticity of $ASIZE_{i,j,t}$ (0.48) showed a positive impact. These two variables can be managed by airlines when they define their level of supply and target market. The variation in the annual means of the variables from 2007 to 2016 shows that the Brazilian airlines are working in the right direction. However, the high level of idle capacity (Table 5) indicates that there is significant room for improvements in operations. It is not only a question of attracting more passengers to air transport, but also of attracting more cargo and postal business. This is not exclusively up to the airlines: the logistical conditions of airport access and bureaucratic streamlining of tax procedures are also extremely important to making this modality of cargo transport more attractive. $IC_{i,j,t}$ is much more elastic than $ASIZE_{i,j,t}$, and accordingly, it should be borne in mind that good productivity can be attained by focusing on $IC_{i,j,t}$, with no need to abandon lower-density regional routes. The idea is to have a balanced fleet and frequency suited to the type of network that is possible for the Brazilian market. Should the fleet development trend observed in Table 2 continue, the only alternative will be to largely abandon the regional routes, which is neither good for Brazil's economic development nor for the future of regular domestic aviation.

Table 10 shows the period fixed effect dummy variable coefficients from 2007 to 2016. They show no trend toward a positive influence in the fuel performance, where four out of ten coefficients are negative. Although the balance is positive, it is a very small positive effect. Considering that, it may be said that technological advancements did not contribute much toward the fuel performance improvement. Only a technological breakthrough in air transport could change that scenario.

**Table 10.** Period fixed effect.

| Year | Effect |
|------|--------|
| 2007 | 0.051331 |
| 2008 | 0.000610 |
| 2009 | −0.044556 |
| 2010 | 0.068665 |
| 2011 | 0.037720 |
| 2012 | 0.005737 |
| 2013 | −0.003723 |
| 2014 | −0.075073 |
| 2015 | −0.030518 |
| 2016 | 0.014526 |

Figure 3 shows the cross-section fixed effect coefficients. From the 988 cross-sections of the study, 550 had a positive dummy coefficient and 438 had a negative one. The dummy coefficients varied from −2.29 to 0.88. Among the 988 cross-sections, 23 had coefficients between −0.50 and −2.29. In contrast, 15 routes had coefficients between 0.50 and 0.88. From this, it is possible to say that only a few routes offered extremely negative or positive operational conditions.

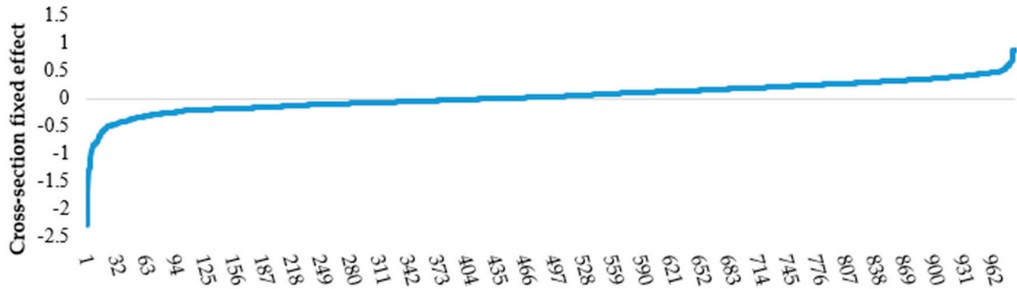

**Figure 3.** Cross-section fixed effect coefficients.

This paper started with the observation of Schnell [3] regarding airlines keeping the excess capacity to cope with future demand or competition. Rationality makes the industry less competitive and inefficient in relation to costs. Once a flight has already occurred, its unused payload represents lost revenue, impacting the company results. This looks like a predation strategy and not a good competitive strategy. Naturally, the reduced supply may lead to customer discomfort, which is another important point in the analysis. However, when considering an activity that is important to a society's economic development but has adverse collateral effects on the environment (which belongs to the whole society), one has to strive for high levels of performance in the use of resources that produce the effect in question. It is important to raise air transport user awareness of the environmental problems connected with this form of transport to raise the tolerance of any possible discomfort that may be necessary in order to increase the productivity of jet fuel burn. In air transport, idle capacity can be divided into two components: the first is where an airline retains capacity that it does not use, such as an aircraft that is idle for lack of demand, while the second is more detrimental, involving non-use of a service, which is the case of unused capacity on a flight. The latter, in addition to generating operating costs with no revenue offset, produces environmental harm. Obviously, it is impossible to eliminate all such idle capacity, but airlines and regulators should seek to develop mechanisms to reduce idle capacity in this industry.

Recent studies have used panel data analysis to find and measure variables related to pollutant emissions, for example, References [24–27]. Aside from this, but aiming toward the same goal, this work analyzed the efficiency of jet fuel use considering its partial productivity (in tonne·km/L). The insight presented here demonstrated that there is another way to reduce air transport pollutant emissions, i.e., by reducing idle capacity.

Another important concern for countries' air transport policy making is the introduction of a pollution tax on airlines, which will inevitably be transferred to the fare prices the airlines charge their customers. In developing countries, such as Brazil, where air transport is under-used and whose transport networks display a number of deficiencies, this subject should be studied with care. If not deployed appropriately, such a tax could be a disincentive to the use of air transport, and by transferring part of the demand to road transport, causing more economic and environmental harm than benefits. Air transport policy should use the tax as an incentive to improve fuel burn productivity by not taxing high-productivity flights, thus encouraging airlines to refine their business portfolios by acting more representatively in cargo transport, for instance. The emphasis on productivity would stimulate airlines to strive for a greater operational efficiency, rather than being discouraged by a market retraction due to higher operating costs.

This discussion indicates that the reduction of idle capacity is the most important operational variable in improving the efficiency of the air fleet in Brazil. A limitation of the study is the standardization that occurred in the Brazilian airline fleet that may not occur in other countries, which gives a certain peculiarity to the case. Park and O'Kelly [28] studied the operating cost of aircraft efficiency in the US aviation market. They concluded that an efficient fleet is composed of mixed-size aircraft as an alternative to a variety of markets. Larger aircrafts tend to be more efficient, using less fuel per passenger [29]. However, for the Brazilian case, the fleet is very similar between airlines, with no large variation in aircraft size, thus in the short term, to measure the efficiency, only the analysis of idle capacity was considered. In an open sky situation, a greater variation of equipment can be observed in the same air route, which can produce different results.

## 5. Conclusions

This paper presented an approach that yielded objective measures for the relation between jet fuel burn, airline operating procedures, and future developments regarding fuel efficiency. It also revealed a convergent association between improved airline operations and finances and reduced aircraft fuel burn, resulting in lower pollutant gas emissions. Using a productivity approach, in physical terms regarding the airlines' main operating cost item, the paper discussed the possibilities of improving the partial productivity of jet fuel for Brazilian regular domestic air transport. In that context, it addressed opportunities for the improvement in technological, regulatory (government), and operational (airline) terms.

Domestic air transport in Brazil offers significant opportunities for improving performance through operational factors. Very high idle capacity can be reduced through operations by making better use of the payload supplied; this improvement can be obtained by developing cargo transport and higher seat-occupation rates. To that end, airlines must manage their service supply and prices more efficiently. Regulatory agencies can contribute significantly by setting standards favoring airlines that achieve lower idle capacity. For example, approval for new frequencies could be made conditional on achieving idle capacity that is more compliant with standards set by the regulator regarding ATK not used per liter of fuel, which is the variable considered for analysis here. The study shows important opportunities for improving airlines' financial performance and making fuel burn more efficient in regular domestic air transport in Brazil. Table 5 shows that although idle capacity is reducing in the regular domestic air transport in Brazil, it is still very high.

Airlines should be encouraged to increase the aircraft size only on routes that can support such growth without a decline in load factor. The study confirms that from both environmental and financial standpoints, it is important that airlines operate with the lowest possible idle capacity, as can be seen from the elasticity performance of this factor in the model shown in Table 9. Airlines' fleet composition is fundamental in order to serve their networks with the lowest possible idle capacity. Increasing aircraft size without prior study and proper regulatory procedures can lead to regional aviation being abandoned or neglected, as has been the trend in Brazilian domestic air transport.

A potential future study would involve a comparison of the fuel efficiency of different airlines operating in developing countries, seeking to identify benchmarks for the airline industry.

**Author Contributions:** Conceptualization, E.F. and R.R.P.; formal analysis, E.F., M.C., P.A., and R.R.P.; funding acquisition, E.F., F.F., M.C., P.A., and R.R.P.; investigation, E.F., M.C., P.A., and R.R.P.; project administration, M.C.; writing—original draft, E.F., M.C., P.A., and R.R.P; writing—review and editing, E.F., F.F., M.C., P.A., and R.R.P. All authors have read and agreed to the published version of the manuscript.

**Funding:** This research received no external funding.

**Acknowledgments:** The authors would like to thank the editor and anonymous reviewers for their insightful comments and suggestions.

**Conflicts of Interest:** The authors declare no conflicts of interest.

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
