# Peer review of "Energy Effectiveness of Jet Fuel Utilization in Brazilian Air Transport"

_sustainability, doi:10.3390/su12010303_

Round 1

Reviewer 1 Report

The proposed manuscript aims to analyze the effectiveness of the jet fuel utilization in the Brazilian air transportation. Many factors are used in the analysis. Unfortunately, the flow of the manuscript is very chaotic. The approach itself is quite interesting, but the description and definition of the problems, aims, assumptions, and methodology need signifficant improvements. 

The structure of the manuscripts need changes.

Generally, the manuscript does not corespond clearly to the scope of the Journal - namely, to the sustainability issues. This is also a weak site of the manuscript.

The introduction should present a state-of -the art. of the disscussed topic. What is done, what is missing etc. Then, the authors must clearly find a gap for their research and the importance of the potential results for science and practise. 

Here, the authors have a kind of the "mess". There is a mix of introduction, literature review and the aims of the mansucript. It must be seperated and puzzled.

Usually, only last paragraph of the introduction presents the aim of the manuscript with short list, what will be invetigated.

Methodology: This is a very important part of the manuscript. Everything must be clearly defined. Methodoly described. All assumptions and equations must be placed here. Also, all data used for the analysis should be presented. All inputs data must be here.

Results: This is the best part of the manuscripts. It is described properly.

Disscusion: This part needs some improvement. The results must be disscussed and compared to the achievements of other researchers.

Conclusions: Among other data, should leave a space for further research and application's potential of the results. 

Detailed remarks:

Title: "Jet Fuel Efficiency in Brazilian Regular Air Transport" - In the title there is a mental shortcut that should be avoided in the scientific papers.

Autors should change the title to avoid confussion, i.e. "Energy effectiveness of the jet fuel utilization in Brazilian Regular Air Transport" - It is just an example

Keywords: some of the keywords written do not correspond to the paper or do not provide expected information. i.e. "environment" - the manuscript does not present any results converted directly to the emissions factors etc. just general context - it is too less. "Panel Data" - it does not provide any information, actually,. 

The manuscript must better puzzled. Many issuses should be combined/collected as they are scattered, i.e. the aims of the research/manuscript or assumptions for the analysis.

- "jet fuel" or "aviation fuel" - please, use only one term in the manuscript.

line 29: "improve aviation fuel efficiency and productivity" - unfortunate statement. it is a mental shortcut. Please, rewrite it. like: improvement of the aviation fuel utilization/usage.

line 30: "Airlines and Turbines Manufactures" - no capital letters. this is not a name of the company or so.

line 38: "[[1]]" - not allowed. correct in the whole paper.

line 68-69: "May we be reaching a limit or is there room to improve performance with existing technologies?" - do not ask general questions if the scientific paper. Rewrite it, please, to pass the message as a unsolved problem or so. 

Line 76-85: "This paper is organized into seven sections.........." this is not a good place for this information. Generally, the auhtors should avoid such statements and organize a manuscript according to the guidelines. It is not a book where in a preface such informations are written. Should be delated.

- the units are missing in the defined symbols of the equations (please, add units

line 274: why 90 kg - the refference in required

line 288: "will be expressed by Equation 4." - rather it is expressed by.... 

line 354: "The minimum flight distance considered for analysis was 200 km" - the refference in required.

line 400: the title of the table should be centredline: "Once a flight is done, its empty payload does no become revenue and it" - it is not well grammatically written

line 469: "This paper presents an" use a past time. the paper is allready written.

Line 494: "it still very high." - should be "it is still ….."

Summarizing, the manuscript must be puzzled, better ordered and presents more sustainable context, as it is a main scope of the Journal.

Reviewer 2 Report

I have reviewed the Manuscript ID: sustainability-667609, with the title "Jet Fuel Efficiency in Brazilian Regular Air Transport". In this paper the authors propose an approach that yields objective measures of the relation between aviation fuel burn, airline operating procedures and future developments regarding fuel efficiency. The authors discuss the productivity of jet fuel in Brazilian domestic aviation and propose an econometric method to support public regulators and airlines decisions.

I consider that the paper will benefit if the authors address within the manuscript the following aspects:

Remarks regarding the sections of the manuscript. In the actual form of the paper, a part of its sections is not according to the ones recommended by the template of the MDPI Sustainability Journal. The manuscript under review will benefit if it is restructured in accordance with the above-mentioned template that provides a more logical structure that is much more appropriate for a research article. The restructuring of the manuscript will also help the authors to express better the novelty of their work and the contribution that they have made to the current state of knowledge. Consequently, the manuscript under review should be restructured as follows: Abstract, Keywords, 1. Introduction, 2. Materials and Methods, 3. Results, 4. Discussion, 5. Conclusions (not mandatory), 6. Patents (not mandatory), Supplementary Materials (not mandatory), Author Contributions, Funding, Acknowledgments, Conflicts of Interest, Appendices and References. Another issue regarding the sections of the manuscript under review is the fact that in its current form there are two sections numbered as "5", namely the sections "5. Case study" and "5. Conclusions". Please address this inconsistency.

Lines 12-25, the Abstract of the paper. It will benefit the paper if in the abstract, in addition to the already presented elements, the authors declare and briefly justify the novelty of their work.

Lines 28-85, the "Introduction" section and Lines 86-161, the "Literature review" section. In the actual form of the manuscript, after the "1. Introduction" section, there exists a section entitled "2. Literature review". I consider that this section's purpose and the one of the "Introduction" are overlapping and therefore I consider that the two sections should be concatenated and reorganized into a single section, namely the "Introduction" (if they consider necessary, the authors can use a subsection "Literature review" within the "Introduction" section).

The "Materials and Methods" section. It will benefit the manuscript if the authors include a "Materials and Methods" section (in the actual form of the manuscript this section is missing, being partially replaced by the section "3. Idle Capacity Analytical Methodology"), in which the new developed methods should be described in detail while well-established methods (and information) can be briefly described and appropriately cited. The authors must restructure their manuscript in order to devise a proper "Materials and Methods" section, eventually structured in subsections. In order to bring a benefit to the manuscript, the authors should state and justify very clear in the "Materials and Methods" section, preferably within the first paragraph, the choices they have made when developing the final form of their proposed approach. The authors should state what has justified using their approach, what is special, unexpected, or different in their research methodology. It will benefit if the authors mention if they have tried other approaches that in the end led them to the current form of their research design.

The "Materials and Methods" section. In order to help the readers better understand the methodology of the conducted study, in the "Materials and Methods" section, the authors should devise a flowchart that depicts the steps that they have processed in developing their research and most important of all, the final target. This flowchart will facilitate the understanding of the proposed approach and it will make the article more interesting to the reader if used as a graphical abstract. This diagram should be analyzed in detail within the manuscript by specifying all the elements needed for each and every step, in order to reach the final result of the study.

In the "Materials and Methods" section, the authors should specify the detailed hardware and software configurations that they have used when developing their research, in order to provide all the necessary details for assuring the reproducibility of the study.

The equations within the manuscript should be explained, demonstrated or cited, as there are some equations that have not been introduced in the literature for the first time by the authors and that are not cited.

Lines 228-229: "The data set was formatted as unbalanced panel data for domestic air routes in Brazil from 2007 to 2016." I would like the authors to comment in the paper whether the data collected during the period 2007-2016 are still relevant today, in 2019, in what concerns the same targeted parameters. The authors should provide explanations whether their study is consistent, whether the changes that may occur within the older dataset from the above-mentioned period and the current year risk altering the final result.

Lines 239-240. "The annual information was organized from the available Brazilian National Civil Aviation Agency (ANAC) data base". The authors must provide more details regarding the way in which they have solved the problems related to missing data or abnormal values if they are to occur.

Lines 331-332, Figure 1. For consistency reasons, in a scientific article one should not represent on the same chart different sets of physical entities. In the case of Figure 1, the authors have represented a first dataset consisting in the number of embarked passengers, a second dataset that consists in the embarked domestic cargo's weight (kg), while the third dataset is being represented by the number of airports. Consequently, the authors should present different charts for the three different sets of entities and adjust the vertical axis accordingly.

The "Discussion" section. In order to validate the usefulness of their research, in the "Discussion" section, the authors should make a comparison between their approach from the manuscript and other similar ones that have been developed in the literature for the same or related purposes. In the actual form of the manuscript, the "Results and Discussion" section contains only one reference to other studies (and even this reference is not being used for comparing the results), so the comparison is missing in the manuscript's current form.

The "Discussion" section. The authors should present the findings and their main implications in the "Discussion" section, also highlighting current limitations of their study, and briefly mention some precise directions that they intend to follow in their future research work.

The "Discussion" section. I consider that the paper will benefit if the authors make a step further, beyond their analysis and provide an insight at the end of the "Discussion" section regarding what they consider to be, based on the obtained results, the most important, appropriate and concrete actions that the decisional factors and all the involved parties should take in order to benefit from the results of the research conducted within the manuscript as to attain the ultimate goal of sustainability.

Issues regarding the format of the paper. The authors must take into account the recommendations from the MDPI Sustainability Journal's website regarding the format of the papers, by using the Microsoft Word template or LaTeX template to prepare their manuscript. In the actual form of the paper, a part of these recommendations has not been taken into account and therefore, the reading of the manuscript is affected, for example:

The citations [1] (line 38), [2] (line 40) and [3] (line 42)  are marked using a double parenthesis system and therefore they are not in accordance to the recommendations of the Sustainability MDPI Journal's Template. According to this template, in the text of the manuscript, the reference numbers should be placed in square brackets [  ] and placed before the punctuation; for example [1], [1–3] or [1,3]. For embedded citations in the text with pagination, use both parentheses and brackets to indicate the reference number and page numbers; for example [5] (p. 10), or [6] (pp. 101–105).

The equations' numbering. At Line 283 appears equation (5), while afterwards, at the Line 289, appears equation (4). The authors must address this inconsistency.

Lines 184-186: "… technological improvement in the industrial sector such as more efficient airplanes and improvements in land operational and management procedures, more direct routes, etc.". In a scientific paper one should avoid using run-on expressions, such as "and so forth", "and so on" or "etc.". Therefore, instead of "etc.", the sentence should mention all the elements that are relevant to the manuscript.

Reviewer 3 Report

The submitted article, Jet Fuel Efficiency in Brazilian Regular Air Transport, by Cabo et al. presents a mathematical/statistical model to study the productivity of jet fuel in Brazilian domestic aviation and recommends an econometric method to support public regulators and airlines decisions.

The proposed model uses variables like aircraft size, route characteristics and idle flight capacity as input parameters for the data analysis using the Chow, Breusch-Pagan and Hausman tests. Reducing idle capacity was found to be the most prominent parameter affecting the short-term fuel efficiency.

Overall, the paper is well-written and nicely presented. I have few minor comments/questions that needs to be addressed before publication-

References for the equations are missing. Please cite the sources. How did you obtain equations (1) and (2). Was the regression analysis performed by authors, or did you adopt the model from somewhere else? Please clarify. If possible, please add the regression plot and highlight the range of regression error. Specify all the assumptions made to derive equations (3)–(6). Few equations seem overly simplified. For instance, fuel productivity ignores the waiting time, which is ok but needs to be mentioned. Conclusions section is too long. I think, part of it can be moved to discussion section. It is not common to have references in the conclusions; it should be just focused on the present work.

Round 2

Reviewer 1 Report

The paper has been significantly improved.

Detailed remarks:

Line 29: is: Air Transport, ;Brazil - should be: Air Transport, Brazil

Line 67: should be “there is a room”

Line 203: is: model Other, should be “model. Other”

Line 351: is “in million(kg);” – should be: “in million (kg);”

Line 453: the caption should be in centre

References should be listened in brackets [2] and [3].

Reviewer 2 Report

I have reviewed the revised version of the Manuscript ID: sustainability-667609, with the title "Jet Fuel Efficiency in Brazilian Regular Air Transport" and, although the authors have modified their paper, there are still many issues that have not been addressed by the authors, even if I have signaled them in my previous review report. In the current form of the paper, the authors have actually procrastinated, postponed some important aspects and consequently, I have made and sent to them the following comments:

The "Materials and Methods" section. In order to help the readers better understand the methodology of the conducted study, in the "Materials and Methods" section, the authors should devise a flowchart that depicts the steps that they have processed in developing their research and most important of all, the final target. This flowchart will facilitate the understanding of the proposed approach and it will make the article more interesting to the reader if used as a graphical abstract. This diagram should be analyzed in detail within the manuscript by specifying all the elements needed for each and every step, in order to reach the final result of the study. Even if I have signaled this issue in my previous review report, the authors did not address it in the revised version of the manuscript.

In the "Materials and Methods" section, the authors should specify the detailed hardware and software configurations that they have used when developing their research, in order to provide all the necessary details for assuring the reproducibility of the study. Even if I have signaled this issue in my previous review report, the authors did not address it in the revised version of the manuscript.

Lines 351-352, Figure 1 (a). For consistency reasons, in a scientific article one should not represent on the same chart different sets of physical entities. In the case of Figure 1 (a), the authors have represented a first dataset consisting in the number of embarked passengers, a second dataset that consists in the embarked domestic cargo's weight (kg). Consequently, the authors should present different charts for the three different sets of entities and adjust the vertical axis accordingly. Even if I have signaled this issue in my previous review report, the authors did not address it in the revised version of the manuscript.

The "Discussion" section. In order to validate the usefulness of their research, in the "Discussion" section, the authors should make a comparison between their approach from the manuscript and other similar ones that have been developed in the literature for the same or related purposes. In the actual form of the manuscript, the "Discussion" section contains only one reference to other studies, [3], and even this reference is not being used for comparing the results, so the comparison is missing in the manuscript's current form. Even if I have signaled this issue in my previous review report, the authors did not address it in the revised version of the manuscript.
